# Diagnostic Value of Dynamic ^18^F-Fluorodeoxyglucose Positron Emission Tomography-Computed Tomography (^18^F-FDG PET-CT) in Cervical Lymph Node Metastasis of Nasopharyngeal Cancer

**DOI:** 10.3390/diagnostics13152530

**Published:** 2023-07-29

**Authors:** Guanglie Li, Shuai Yang, Siyang Wang, Renwei Jiang, Xiwei Xu

**Affiliations:** 1Department of Head and Neck Oncology, The Fifth Hospital of Sun Yat-sen University, Zhuhai 519000, China; liglie@mail.sysu.edu.cn (G.L.); 13570608929@163.com (S.W.); 2Department of Radiotherapy Physics, The Fifth Hospital of Sun Yat-sen University, Zhuhai 519000, China; yangsh67@mail.sysu.edu.cn

**Keywords:** dynamic PET-CT, SUV-max, K_i_-mean, K_i_-max, nasopharyngeal cancer, cervical lymph nodes

## Abstract

Background and purpose: Dynamic ^18^F-FDG PET-CT scanning can accurately quantify ^18^F-FDG uptake and has been successfully applied in diagnosing and evaluating therapeutic effects in various malignant tumors. There is no conclusion as to whether it can accurately distinguish benign and malignant lymph nodes in nasopharyngeal cancer. The main purpose of this study is to reveal the diagnostic value of dynamic PET-CT in cervical lymph node metastasis of nasopharyngeal cancer through analysis. Method: We first searched for cervical lymph nodes interested in static PET-CT, measured their SUV-Max values, and found the corresponding lymph nodes in magnetic resonance images before and after treatment. The valid or invalid groups were included according to the changes in lymph node size before and after treatment. If the change in the product of the maximum diameter and maximum vertical transverse diameter of the lymph node before and after treatment was greater than or equal to 50%, they would be included in the valid group. If the change was less than 50%, they would be included in the invalid group. Their K_i_ values were measured on dynamic PET-CT and compared under different conditions. Then, we conducted a correlation analysis between various factors and K_i_ values. Finally, diagnostic tests were conducted to compare the sensitivity and specificity of K_i_ and SUV-Max. Result: We included 67 cervical lymph nodes from different regions of 51 nasopharyngeal cancer patients and divided them into valid and invalid groups based on changes before treatment. The valid group included 50 lymph nodes, while the invalid group included 17. There wer significant differences (*p* < 0.001) between the valid and the invalid groups in SUV-Max, K_i_-Mean, and K_i_-Max values. When the SUV-Max was ≤4.5, there was no significant difference in the K_i_-Mean and K_i_-Max between the two groups (*p* > 0.05). When the SUV-Max was ≤4.5 and pre-treatment lymph nodes were <1.0 cm, the valid group had significantly higher K_i_-Mean (0.00910) and K_i_-Maximum (0.01004) values than the invalid group (K_i_-Mean = 0.00716, K_i_-Max = 0.00767) (*p* < 0.05). When the SUV-Max was ≤4.5, the pre-treatment lymph nodes < 1.0 cm, and the EBV DNA replication normal, K_i_-Mean (0.01060) and K_i_-Max (0.01149) in the valid group were still significantly higher than the invalid group (K_i_-Mean = 0.00670, K_i_-Max = 0.00719) (*p* < 0.05). The correlation analysis between different factors (SUV-Max, T-stage, normal EB virus DNA replication, age, and pre-treatment lymph node < 1.0 cm) and the K_i_ value showed that SUV-Max and a pre-treatment lymph node < 1.0 cm were related to K_i_-Mean and K_i_-Max. Diagnostic testing was conducted; the AUC value of the SUV-Max value was 0.8259 (95% confidence interval: 0.7296–0.9222), the AUC value of the K_i_-Mean was 0.8759 (95% confidence interval: 0.7950–0.9567), and the AUC value of the K_i_-Max was 0.8859 (95% confidence interval: 0.8089–0.9629). After comparison, it was found that there was no significant difference in AUC values between K_i_-Mean and SUV-Max (*p* = 0.220 > 0.05), and there was also no significant difference in AUC values between K_i_ max and SUV-Max (*p* = 0.159 > 0.05). By calculating the Youden index, we identified the optimal cut-off value. It was found that the sensitivity of SUV-Max was 100% and the specificity was 66%, the sensitivity of K_i_-Mean was 100% and the specificity was 70%, and the sensitivity of K_i_-Max was 100% and the specificity was 72%. After Chi-Square analysis, it was found that there was no significant difference in specificity between K_i_-Mean and SUV-Max (*p* = 0.712), and there was also no significant difference in specificity between K_i_-Max and SUV-Max (*p* = 0.755). Conclusion: Dynamic PET-CT has shown a significant diagnostic value in diagnosing cervical lymph node metastasis of nasopharyngeal cancer, especially for the small SUV value, and lymph nodes do not meet the metastasis criteria before treatment, and EBV DNA replication is normal. Although the diagnostic accuracy, sensitivity, and specificity of dynamic PET-CT were not significantly different from traditional static PET-CT, the dynamic PET-CT had a more accurate tendency.

## 1. Introduction

Nasopharyngeal cancer is a malignant tumor originating from nasopharyngeal mucosal epithelial cells. It is prevalent in southern China and Southeast Asia, and its incidence rate is 20–30/10,000/year [1]. Nasopharyngeal cancer has a hidden onset, and the early symptoms are not obvious. When apparent symptoms appear, it is already in the middle and late stages, and most patients only want to come to the hospital for the first time when they have touched the noticeable neck lumps [2]. It is essential to determine whether cervical lymph nodes are metastatic for nasopharyngeal cancer patients to clarify clinical staging, formulate treatment plans, and evaluate prognosis. Unlike other head and neck malignant tumors, nasopharyngeal cancer mainly adopts a comprehensive treatment plan based on radiation therapy, with local recurrence being the leading cause of treatment failure. Among them, 14–18% are cervical lymph node recurrence [3], mainly because the missed false negative lymph nodes were classified as low-dose areas during the planning process.

Previous studies have reported that ^18^F-fluorodeoxyglucose positron emission tomography-computed tomography (^18^F-FDG PET-CT) has shown significant functionality in detecting distant metastasis of nasopharyngeal cancer [4,5]. ^18^F-FDG PET-CT scanning is the most commonly used imaging method in clinical practice, which can reflect prognostic factors directly related to clinical outcomes, including maximum standardized uptake value (SUV-Max), metabolic tumor volume, and total lesion glycolysis [6]. ^18^FDG PET-CT combines the anatomical accuracy of CT and the details of PET molecular metabolism and has become an imaging method for molecular targets, providing an important basis for differential diagnosis and therapeutic evaluation of tumors [7]. However, static ^18^F-FDG PET-CT only displays the cumulative results after 60 min of tracer injection, ignoring the intermediate processes, including the transport of the tracer through blood flow, its exchange between blood vessels and tissues, and its binding and separation with the target [8]. SUV is an important diagnostic indicator for static PET imaging, a standard for given drug activity and weight, and a relative measurement value [9,10] and is influenced by many factors related to physical effects, hardware and software system specifications, tracer kinetics, motion, scanning protocol design, and limitations of current image derived PET indicators [11]. Meanwhile, the average SUV value changes due to voxel levels, making it sensitive to the definition of ROI (region of interest) and influenced by changes within and between observers [12]. Dynamic PET-CT scanning refers to starting data collection immediately after injection of a tracer, generating a time-activity-curve (TAC) based on the framing during the scanning process, and extracting parameters such as K_1_-K_4_, K_i_, and glucose metabolism rate through dynamic modeling. It can avoid the influence of factors such as uptake kinetics, injection imaging time, and BMI, has better accuracy than SUVs, and can achieve a quantitative evaluation of tumor metabolism. In addition, dynamic studies have found that K_i_ can more sensitively identify early metastatic lymph nodes than SUVs [13]. For suspected metastases that SUVs cannot accurately identify, dynamic PET-CT scans can provide more information.

Recently, multiple studies have confirmed the important role of dynamic PET-CT in determining lymph node metastasis. Sun [14] et al. conducted a prospective cohort study on patients with non-small cell lung cancer and obtained some characteristic dynamic parameters such as V_Median, K_3__Entropy, VB_Entropy, K_1__Uniformity, and K_i__Uniformity. The results showed significant differences between benign and malignant lymph nodes, which were pathologically confirmed. The results showed significant differences in K_3__Entropy, VB_Entropy, K_1__Uniformity, and K_i__Uniformity. The K_i_, VB, and K_3_ regression models could make good predictions for distinguishing between benign and malignant lymph nodes. Wumner [15] et al. analyzed 135 mediastinal and hilar lymph nodes from 29 lung cancer patients and confirmed pathologically that 49 were non-metastatic and 86 were metastatic. All patients completed dynamic PET-CT and obtained characteristic parameters. In addition to K_3_, dynamic metabolic parameters K_1_, K_2_, K_i_, and K_i_/K_1_ also performed well in the differential diagnosis of metastatic and non-metastatic lymph nodes (*p* < 0.05). However, the predictive role of dynamic PET-CT in cervical lymph node metastasis of nasopharyngeal carcinoma has not been reported.

Therefore, the primary purpose of this study is to clarify whether the K_i_ value in dynamic PET-CT can replace the SUV value in distinguishing cervical lymph node metastasis of nasopharyngeal cancer, which can be used as a reference for radiation oncologists when delineating the target area, and can improve tumor control rate, reduce recurrence rate, and obtain better efficacy and prognosis in clinical treatment.

## 2. Material and Method

### 2.1. Patients

A retrospective analysis was conducted on 51 newly diagnosed nasopharyngeal cancer patients who visited our department from 2020 to 2022. After the precise diagnosis, all patients received systematic treatment (radiotherapy, chemotherapy, or other anti-tumor treatments). All patients met the following criteria: (1) The pathological diagnosis was confirmed as nasopharyngeal cancer after the nasal endoscopic biopsy or multiple nasal endoscopic examinations could not diagnose nasopharyngeal cancer, but cervical lymph node biopsy was confirmed as metastatic squamous cell carcinoma and considered as the nasopharyngeal source, and there is evidence of nasopharyngeal malignancy on imaging. (2) No anti-tumor treatment was initiated before the PET-CT examination. (3) Before and after treatment, magnetic resonance imaging of the head and neck was performed, and there were measurable target lesions. (4) No other malignant tumors were present. (5) There was no severe cardio-cerebrovascular disease or diabetes, or mental disease.

### 2.2. Data Acquisition and Image Reconstruction

The patients fasted for at least 6 h before the examination, and their blood sugar was controlled at the normal level before the examination. Then, a uMI780 PET/CT scanner (Shanghai United Imaging Company, Shanghai, China) was used for scanning. Each patient underwent the low-dose transmission CT scan of the whole body, with a tube voltage of 120 KV and a tube current of 180 mA. The focus was on observing primary nasopharyngeal tumors and cervical lymph nodes. Then, immediately after injection of ^18^F-FDG (3.7 MBq/kg), a 60 min single bed dynamic PET image was collected, and an 18 × 5 s, 6 × 10 s, 5 × 30 s, 5 × 60 s, 8 × 150 s, 6 × 300 s frame protocol was divided into 48 image data [13]. Then, we used four beds (2 min each) for routine static whole-body PET scans and used the ordered subset expectation maximization (OSEM) iterative algorithm (parameter: 2 subsets, 20 iterations, 150 × 150 matrices) and reconstructed all dynamic PET data. Similarly, conventional static SUV images were reconstructed using the OSEM method (parameters: 2 subsets, 20 iterations, 128 × 128 matrices). All data were corrected according to isotope decay, scattering events, and random coincidence, and semi-maximum smoothing was performed using a standard Gaussian filter with a total width of 3 mm.

Our research used internally developed computer code for indirect parameter reconstruction processing [16,17]. Assuming irreversible uptake of ^18^F-FDG, the physiological parameters of voxel levels were estimated using the ordinary least squares (OLS) Patlak reconstruction regression method for 48 dynamic PET series based on a dual tissue compartment dynamics model [18]. We estimated the tracer concentration C_p_(t) in the bleeding plasma and the tracer concentration C(t) in the tissue from the PET-CT images. In this study, we defined the input function C_p_(t), derived non-invasively from dynamic PET data as the active concentration of the tracer in plasma. Specifically, VOI was drawn on the aortic arch of each patient, and then input functions derived from arterial images were extracted from all dynamic PET sequences. Then, by applying the OLS linear regression method to the following Patlak equation, the tracer uptake rate K_i_ at the voxel level could be calculated [18,19,20,21]. The specific block diagram was shown in Figure 1, and its equation was as follows: K_i_-Mean was defined as the average K_i_ value at which the tracer was uptaken, and K_i_-Max was the K_i_ value at which the uptake rate reached the highest value.
$$\frac{C\left(t\right)}{C_{p}\left(t\right)}=K_{i}\frac{\int_{0}^{\tau }C_{P}\left(\tau \right)d\tau }{C_{p}\left(t\right)}+V,t\geq t^{*}{\;\;K}_{i}=\frac{K_{1}\cdot K_{3}}{K_{2}+K_{3}}$$
where t was the time when the vascular space and reversible tissue compartment reached relative equilibrium and K_i_ and V were the slopes and intercepts of linear regression, respectively.

### 2.3. Delineation of VOIs

The delineation of VOIs (volumes of interest) mainly included cervical lymph nodes. Two experienced radiology doctors (Yang and Jiang) initially determined whether it was metastasis or reactive hyperplasia based on different lymph nodes’ size, shape, and SUV values in static PET-CT images and then measured the maximum SUV value (SUV-Max) of these suspicious lymph nodes and incorporated them into the following dynamic PET-CT image reconstruction software for analysis. Subsequently, the specific size changes of these lymph nodes before and after treatment were analyzed through comparison in magnetic resonance images, and data were recorded. The delineation of VOIs was performed by two radiation oncologists (Li and Xu). The K_i_ values of these lymph nodes were measured using Carisma software (version 2.0), which was used for comprehensive analysis with SUV values and changes in size before and after treatment.

### 2.4. Efficacy Evaluation

Due to the inability to perform needle biopsies on all cervical lymph nodes of all patients, we performed needle biopsies to determine the benign and malignant nature of cervical lymph nodes only when the patient had the following conditions, such as patients had obvious cervical masses, suspected metastatic lymph nodes on imaging, or multiple biopsies from the nasopharynx could not determine the diagnosis. Only a few patients underwent needle biopsies of cervical lymph nodes. For most patients, we mainly identified the possibility of malignancy through changes in size on MRI before and after treatment. According to the World Health Organization (WHO) efficacy evaluation criteria, clinical efficacy was evaluated [22]. Based on the definitions of partial remission (PR) and complete remission (CR), we considered these two situations more likely for the lymph node to be malignant. That is to say, we supposed that the product value of the lymph node’s maximum diameter and maximum vertical transverse diameter changed more than or equal to 50% before and after treatment. In that case, treatment was considered valid, indicating a tendency towards malignancy. Treatment was considered invalid if it was less than 50%, indicating a greater likelihood of benign.

### 2.5. Data Analysis

Two radiation oncologists (Li and Xu) accurately recorded all data, performed the normality distribution test on all data, and used Pearson or Spearman tests to analyze the correlation between different factors and K_i_ values based on the normal distribution. For sensitivity and specificity detection of diagnosis, we drew ROC curves, calculated the area under the curve, and found the optimal cutoff point. If the data followed the normal distribution, the *t*-test was used to analyze the K_i_ values under different conditions (SUV-Max value, lymph node size before and after treatment, and EB virus DNA expression). Conversely, the Mann–Whitney U test was used, and all tests were bilateral tests, with the *p*-value < 0.05 indicating meaningful results. All statistical analyses were plotted and analyzed using GraphPad PRISM version 9.0 and SPSS software version 27.0.

## 3. Result

### 3.1. Patients

Ultimately, we screened 51 patients with confirmed pathological diagnoses of nasopharyngeal cancer, including 36 male and 15 female patients. All patients received comprehensive treatment, such as radiotherapy and chemotherapy. All patients underwent MRI examinations before and after treatment, and some patients underwent cervical lymph node biopsy before treatment, and their benign or malignant nature was consistent with the degree of change in the MRI before and after treatment. The clinical characteristics of these 51 patients are shown in Table 1. The median age of all patients was 48 years (range: 26–63). In TNM staging, there were 7 patients with T1, 9 with T2, 28 with T3, and 7 with T4. There were 21 patients with stage N1, 27 with stage N2, and 3 with stage N3. There were 3 patients with stage M0, 44 with stage M0, and 4 with stage M1. Among all clinical stages, there were no patients belonging to stage I, while there were 8 patients in stage II, 27 in stage III, and 42 in stage IV. One patient had the TNM stage of T3N1Mx, and it was impossible to determine whether the clinical stage was stage III or IV. Therefore, the patient was included in the unclear staging group. After grouping based on the expression of EBV-DNA, it was found that 23 patients had higher than normal expression levels, while 28 patients had normal expression levels.

### 3.2. Comparison of SUVmax, K_i_-Mean, and K_i_-Max Values between the Valid and Invalid Groups

We identified 67 cervical lymph nodes from different regions from the PET CT images of the 51 patients, measured their static SUV-Max values and dynamic K_i_-Mean and K_i_-Max values, and combined them with magnetic resonance images before and after treatment to determine their size changes. Finally, they were included in the valid and invalid groups based on the degree of change. After the normal distribution test, it was found that the data did not conform to the normal distribution, so the Mann–Whitney U test was used for all comparative analyses.

From Table 2 and Figure 2, it could be seen that the average SUV-Max value of the valid group was 7.2, the average K_i_-Mean was 0.01323, and the average K_i_-Max was 0.01510, which were significantly higher than the invalid group (SUV-Max = 4.3, K_i_-Mean = 0.00978, and K_i_-Max value = 0.01077, *p* < 0.001) (Table 2 and Figure 1).

### 3.3. Comparison of K_i_-Mean and K_i_-Max Values between Valid and Invalid Groups When SUV-Max ≤ 4.5

Matsubara [23] found in their study that when the SUV-Max of cervical lymph nodes was greater than 4.5, it was confirmed by biopsy pathology as metastatic lymph nodes. However, for lymph nodes with an SUV-Max less than or equal to 4.5, it was difficult to distinguish whether they were malignant or benign. Based on the results of this study, we set the case where the SUV-Max was less than or equal to 4.5 in this study. The results showed that there was no significant difference between the K_i_-Mean (*p* = 0.151 > 0.05) and K_i_-Max (*p* = 0.075 > 0.05) between the two groups (Table 3 and Figure 3).

### 3.4. Comparison of K_i_-Mean and K_i_-Max between the Valid and Invalid Groups with an SUV-Max ≤ 4.5 and Lymph Nodes < 1.0 cm before Treatment

The current guidelines and consensus often use a cervical lymph node short longitude ≥ 1.0 cm as a standard for lymph node metastasis, and it is necessary to combine the shape, density, number, and other factors of lymph nodes to assist in diagnosis and treatment [24]. The role of dynamic PET-CT in distinguishing benign and malignant cervical lymph nodes is not yet clear when the cervical lymph node is <1.0 cm. In this study, two conditions were set: (1) SUV-Max was less than or equal to 4.5, (2) cervical lymph node size was <1.0 cm. After analysis, it was found that the K_i_-Mean (0.00910) and K_i_-Max (0.01004) of the valid group were both higher than those of the invalid group (K_i_-Mean = 0.00716 and K_i_-Max = 0.00767, *p* < 0.05) (Table 4 and Figure 4).

We selected two patients who underwent dynamic and static PET/CT scans and cervical lymph node biopsies before treatment and completed MRI examinations before and after treatment. The relevant results were shown in Figure 5. These two patients’ cervical lymph nodes were <1.0 cm in size before treatment, and both showed low metabolism (SUV-Max = 1.6) on static PET-CT. Without biopsies, their benign and malignant status could not be determined, and from the results in Figure 4, one of the individuals had a positive cervical lymph node biopsy result. From the MRI images before and after treatment, there was a significant change in the size of the cervical lymph node (>50%). The K_i_-Mean was 0.01065, the K_i_-Max was 0.01156, and the metabolism was elevated in dynamic PET CT images. On the contrary, the biopsy result of another patient was negative, and there was almost no change in the cervical lymph node before and after treatment. The K_i_-Mean was 0.00658, the K_i_-Max was 0.00749, and the metabolism was not obvious in dynamic PET-CT images. The K_i_ values of the malignant lymph node were significantly greater than that of the benign lymph node. This result indirectly indicated that dynamic PET-CT could be the best choice for low SUV values and sizes < 1.0 cm, which could not determine the nature of cervical lymph nodes.

### 3.5. Comparison of K_i_-Mean and K_i_-Max between the Valid and Invalid Groups with an SUV-Max ≤ 4.5 and a Lymph Node < 1.0 cm and a Normal EBV-DNA Replication before Treatment

Epstein–Barr virus DNA (EBV-DNA) is often used as an important indicator for efficacy monitoring [25,26], tumor recurrence [27], and lymph node metastasis [28] in clinical diagnosis and treatment. If the EBV-DNA level continues to rise, it often indicates a poor prognosis and the possibility of cervical lymph node metastasis. We continued to add EBV-DNA conditions based on an SUV-Max less than or equal to 4.5 and a cervical lymph node less than 1.0 cm before treatment. When EBV-DNA replication was at the normal level, the K_i_-Mean (0.01060) and K_i_-Max (0.01149) of the valid group were both more significant than those of the invalid group (K_i_-Mean = 0.00670 and K_i_-Max = 0.00719, *p* < 0.01) (Table 5 and Figure 6).

### 3.6. The Correlation between Different Factors and K_i_-Mean and K_i_-Max

We conducted a correlation study to explore the correlation between different factors (SUV-Max, T-stage, normal EBV-DNA replication, age, and a cervical lymph node < 1.0 cm) and K_i_-Mean and K_i_-Max values. Because the data did not conform to the normal distribution, we used the Spearman test to determine correlations. The results showed the correlation between SUV-Max and a cervical lymph node < 1.0 cm and K_i_-Mean and K_i_-Max (*p* < 0.0001). The r coefficient of the SUV Max and K_i_-Mean was 0.8450, and the r coefficient of the K_i_-Max was 0.8498. There were significant positive correlations (Table 6 and Figure 7).

### 3.7. Diagnostic Accuracy of SUV-Max and K_i_ for Cervical Lymph Node Metastasis in Nasopharyngeal Cancer

We plotted ROC curves to evaluate the diagnostic accuracy of SUV-Max and K_i_ for cervical lymph node metastasis. The AUC value of the SUV-Max value was 0.8259 (95% confidence interval: 0.7296–0.9222), the AUC value of the K_i_-Mean was 0.8759 (95% confidence interval: 0.7950–0.9567), and the AUC value of the K_i_-Max was 0.8859 (95% confidence interval: 0.8089–0.9629). After comparison, it was found that there was no significant difference in AUC values between K_i_-Mean and SUV-Max (*p* = 0.220 > 0.05), and there was also no significant difference in AUC values between K_i_-Max and SUV-Max (*p* = 0.159 > 0.05). To calculate the sensitivity and specificity of each index, we first calculated Youden’s index. The calculation formula was: Youden’s index = Sensitivity + Specificity − 1. When Youden’s index was the largest, its cut-off value was the best [29]. After calculation, the sensitivity of SUV-Max was 100% and the specificity was 66%. The sensitivity of the K_i_-Mean was 100% and the specificity was 70%; the sensitivity of the K_i_-Max was 100% and the specificity was 72%. After Chi-Square analysis, it was found that there was no significant difference in specificity between K_i_-Mean and SUV-Max (*p* = 0.712), and there was also no significant difference in specificity between K_i_-Max and SUV-Max (*p* = 0.755) (Figure 8).

## 4. Discussion

In the first diagnosis, 70~80% of nasopharyngeal cancer patients have enlarged cervical lymph nodes [30]. The location of metastatic lymph nodes is related to the lymphatic drainage area of the primary tumor. The cervical lymph node metastasis of nasopharyngeal cancer mainly occurs bilaterally, commonly around the jugular vein chain. Currently, the treatment of nasopharyngeal cancer mainly relies on radiation therapy as a comprehensive treatment. With the development of radiation therapy technology, although intensity-modulated radiation therapy (IMRT) can provide excellent dose coverage for tumors, better protect surrounding normal tissues, improve local control, and long-term survival [31,32], 7% to 18% of patients still have residual or recurrent cervical lymph nodes after the first round of radiation therapy [33,34]. Therefore, determining whether neck lymph nodes are metastatic is crucial for accurate staging, selection of treatment plans, delineation of radiotherapy targets, and evaluation of the prognosis of nasopharyngeal cancer.

PET imaging for cancer metabolic assessment has been widely used in clinical medicine. The commonly used method is to evaluate the energy consumption of tumors by injecting ^18^F-fluorodeoxyglucose and detecting the maximum glucose metabolism (SUV-Max) [35]. However, many factors, such as uptake kinetics, body mass index, or post-injection time, can affect the results of SUV values [36]. As a new imaging technology in nuclear medicine, dynamic PET-CT images collect continuous frames through long-term scanning compared with static PET-CT images. Therefore, the degree of drug metabolism and histopathological activity is dynamically reflected [35]. The diagnostic value of dynamic PET-CT has been supported by data in non-small cell lung cancer and primary tumors of nasopharyngeal cancer [13,37], but there is no research report on whether it can differentiate cervical lymph node metastasis in nasopharyngeal cancer.

In our study, we first searched for cervical lymph nodes of interest on PET-CT images and then found the corresponding lymph nodes on magnetic resonance images. Then, according to the latest version of WHO solid tumor evaluation standards, we measured the maximum diameters and maximum vertical transverse diameters of lymph nodes before and after treatment and calculated the product size of the two. If the change before and after treatment exceeded 50%, it was included in the valid group; if it was less than 50%, it was included in the invalid group. Due to the inability to perform a needle biopsy on each cervical lymph node to determine its nature, we could only speculate on its potential for malignancy based on changes before and after treatment. It would likely be malignant if the change was greater than 50%. If the change was less than 50%, it was considered more likely to be benign. Through analysis, we found that the SUV-Max, K_i_-Mean, and K_i_-Max of the valid group were significantly higher than those of the invalid group (*p* < 0.001). This result might indicate that glucose metabolism’s degree and metabolic rate in general metastatic lymph nodes are significantly higher than in benign lymph nodes.

Previous studies have shown that when the SUV value of the lymph node is greater than 4.5, pathological biopsy confirms it as metastasis. However, for SUV values ≤ 4.5, it is difficult to distinguish between benign and malignant [23,38]. Given this result, we also set the same conditions in this study and found no significant difference in the K_i_-Mean and K_i_-Max between the two groups (*p* > 0.05). However, it could be seen from the results that the *p*-value of the K_i_-Max was very close to 0.05. Due to the small sample size of this study, no positive results were obtained. Increasing the sample size further might result in better research results. According to the current results, when the SUV-Max was ≤4.5, the K_i_-Mean and K_i_-Max could not effectively distinguish between metastatic or benign lymph nodes.

The lymph node size is often an important indicator for diagnosing metastasis. In clinical practice, a cervical lymph node size ≥ 1.0 cm is an important criterion for cervical lymph node metastasis. For lymph nodes < 1.0 cm, it is often necessary to consider specific situations, such as whether there is central necrosis, cluster distribution, obvious enhancement, and so on. In our study, it was found that when the SUV-Max was ≤4.5, and the cervical lymph node < 1.0 cm before treatment, it could be seen that the K_i_-Mean and K_i_-Max of the valid group were greater than those of the invalid group (*p* < 0.05). The average K_i_-Mean and K_i_-Max values of the valid group were 0.00910 and 0.01004, and both were close to 0.01. This result indicates that dynamic PET-CT can often distinguish between benign and malignant lymph nodes when the SUV-Max is small and the cervical lymph node size does not meet the standards.

Epstein–Barr (EB) is the γ herpesvirus [39] and is closely related to the occurrence and development of nasopharyngeal cancer. In recent years, research has found that EB virus DNA (EBV-DNA) plays an important role in the efficacy monitoring and prognosis evaluation of nasopharyngeal cancer patients, especially in the clinical significance of changes in EBV-DNA concentration before and after treatment for distant metastasis and local recurrence in nasopharyngeal cancer patients [40]. Current research shows a positive correlation between EBV-DNA content in the blood of nasopharyngeal cancer patients and the volume of cervical lymph node metastasis [41], and EBV is closely associated with lymph node metastasis in nasopharyngeal cancer [28]. To further explore whether dynamic PET-CT can identify cervical lymph node metastasis when the level of EBV-DNA replication was normal, we analyzed it. The results showed that when the SUV-Max was ≤4.5, the pre-treatment lymph node < 1.0 cm, and EBV-DNA replication was normal the K_i_-Mean and K_i_-Max in the valid group were significantly higher than those in the invalid group (*p* < 0.05). Moreover, the valid group’s K_i_-Mean and K_i_-Max were greater than 0.01. Dynamic PET-CT can identify metastatic cervical lymph nodes when the EBV-DNA replication is normal, meanwhile, the SUV-Max is small and the lymph nodes do not meet the standards.

Many factors, such as age, T-stage, and SUV-Max, are related to parameters in dynamic PET-CT. Through analysis, it was found that SUV-Max and pre-treatment lymph nodes < 1.0 cm were associated with K_i_-Mean and K_i_-Max. A strong linear correlation existed between SUV-Max and K_i_-Mean and K_i_-Max. After further diagnostic testing, it was found that the AUC value of both the K_i_-Mean and K_i_-Max were greater than those of the SUV-Max. Generally speaking, the larger the AUC value, the higher its diagnostic or exclusion value [42]. However, in our study, it was found that there was no significant difference in AUC values between K_i_-Mean and SUV-Max (*p* = 0.220 > 0.05), and there was also no significant difference in AUC values between K_i_-Max and SUV-Max (*p* = 0.159 > 0.05). In terms of sensitivity, the sensitivity of SUV-Max, K_i_-Mean, and K_i_-Max could all reach 100%. However, regarding specificity, there was no significant difference between K_i_-Mean and K_i_-Max compared to SUV-Max (*p* > 0.05). The above results may indicate that dynamic PET-CT has the more accurate tendency, and due to sample size limitations, no positive conclusion has been drawn. This positive conclusion may be confirmed in large sample data in the future.

Some issues need to be addressed in our research. (1) We could only speculate on the possibility of cervical lymph node metastasis based on the changes in MRI before and after treatment, without pathological results to support it. There might be misdiagnoses and errors in the conclusions. (2) Some studies yielded nearly positive results, but a positive conclusion could not be reached due to insufficient sample size and could only be treated as negative. (3) In our study, some nasopharyngeal cancer patients only received partial treatment, and the follow-up time after treatment was insufficient. Some cervical lymph nodes might not have shown significant changes and had been included in the invalid group. (4) The sample difference between the valid and invalid groups was large, and the data did not conform to the normal distribution, affecting the results’ authenticity.

## 5. Conclusions

Compared with traditional static PET-CT, dynamic PET-CT has shown advantages in the diagnosis of cervical metastasis of nasopharyngeal cancer, especially for the cervical lymph nodes with small SUV values on static PET-CT, the lymph nodes’ size does not meet the criteria for judging metastasis, and the patient’s serum EBV-DNA replication before treatment is normal. Although our diagnostic test did not yield positive results, it could be seen from this result that dynamic PET-CT has the more accurate tendency. In the future, we need to combine pathology further and increase the sample size to verify everything based on this study.

## Figures and Tables

**Figure 1 diagnostics-13-02530-f001:**
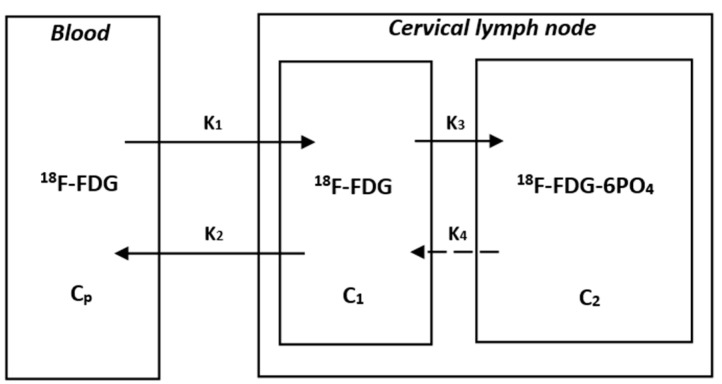
The Patlak model of ^18^F-FDG catabolism. C_P_ is the active concentration of 18F-FDG in plasma, while C_1_ and C_2_ are the concentrations of ^18^F-FDG-6PO_4_ in cervical lymph nodes. K_1_, K_2_, K_3_, and K_4_ are rate constants. K_1_ is the rate constant of blood to the cervical lymph node. K_2_ is the rate constant of the cervical lymph node to blood. K_3_ is the rate constant of ^18^F-FDG to ^18^F-FDG-6PO_4_. K_4_ is the rate constant of ^18^F-FDG-6PO_4_ to ^18^F-FDG. Due to the assumption of a unidirectional uptake of ^18^F-FDG in the Patlak analysis, the rate constant of ^18^F-FDG-6PO_4_ to ^18^F-FDG is negligible. The slope (K_i_) and intercept (V) can be obtained from the graphical analysis and equation, where the slope K_i_ is equal to K_1_K_3_/(K_2_ + K_3_).

**Figure 2 diagnostics-13-02530-f002:**
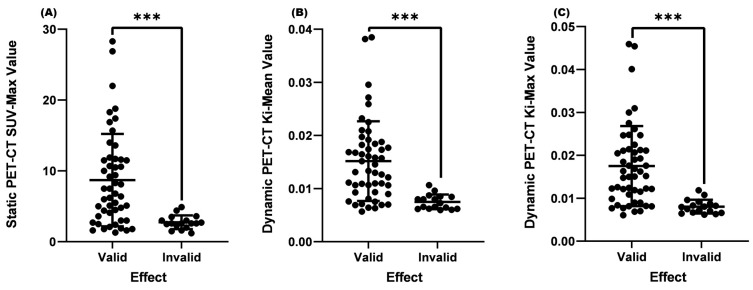
Comparison of SUV-Max, K_i_-Mean, and K_i_-Max between the valid and invalid groups. (**A**) Comparison of SUV-Max between the valid and invalid groups (*p* < 0.001). (**B**) Comparison of the K_i_-Mean between the valid and invalid groups (*p* < 0.001). (**C**) Comparison of the valid K_i_-Max between the valid and invalid groups (*p* < 0.001). *** means *p* < 0.001.

**Figure 3 diagnostics-13-02530-f003:**
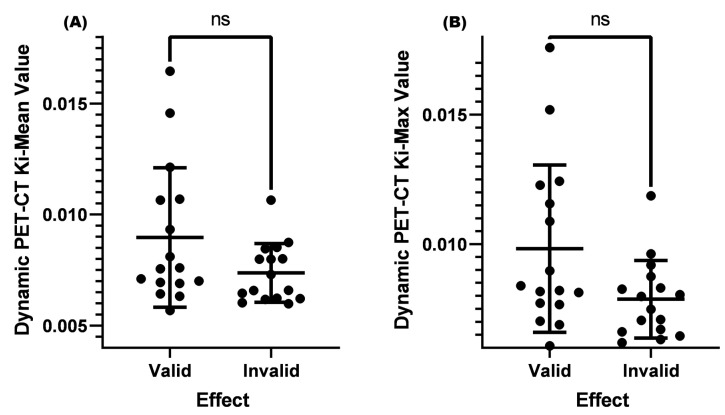
Comparison of K_i_-Mean and K_i_-Max values between valid and invalid groups when the SUV-Max was ≤4.5. (**A**) Comparison of the K_i_-Mean between the valid and invalid groups when the SUV-Max was ≤4.5 (*p* = 0.151 > 0.05). (**B**) Comparison of the K_i_-Max between the valid and invalid groups when the SUV-Max was ≤4.5 (*p* = 0.075 > 0.05). ns means not significant.

**Figure 4 diagnostics-13-02530-f004:**
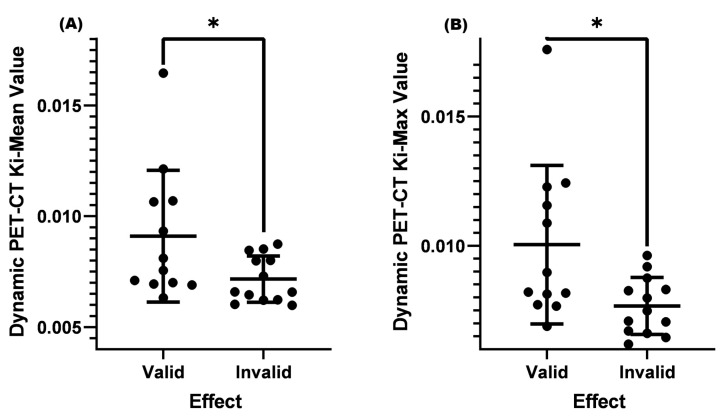
Comparison of K_i_-Mean and K_i_-Max values between the valid and invalid groups with an SUV-Max ≤ 4.5 and a lymph node < 1.0 cm before treatment. (**A**) Comparison of K_i_-Mean between the valid and invalid groups when the SUV-Max was ≤4.5 and the cervical lymph node < 1.0 cm before treatment (*p* = 0.0457 < 0.05). (**B**) Comparison of K_i_-Max between the valid and invalid groups when the SUV was ≤4.5 and the cervical lymph node < 1.0 cm before treatment (*p* = 0.0298 < 0.05). * means *p* < 0.05.

**Figure 5 diagnostics-13-02530-f005:**
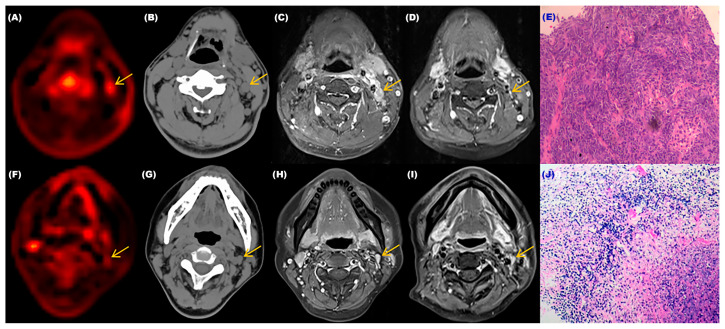
The relevant examination results of two patients with different cervical lymph nodes. (**A**) Dynamic PET-CT image of the patient with the malignant cervical lymph node. (**B**) CT image of the patient with the malignant cervical lymph node. (**C**,**D**) MRI images of the patient with the malignant cervical lymph node before and after treatment. (**E**) Pathological image of the patient’s lymph node which was confirmed as malignant. (**F**) Dynamic PET-CT image of the patient with the benign cervical lymph node. (**G**) CT image of the patient with the benign cervical lymph node. (**H**,**I**) MRI images of the patient with the benign cervical lymph node before and after treatment. (**J**) Pathological image of the patient’s lymph node that was confirmed as benign.

**Figure 6 diagnostics-13-02530-f006:**
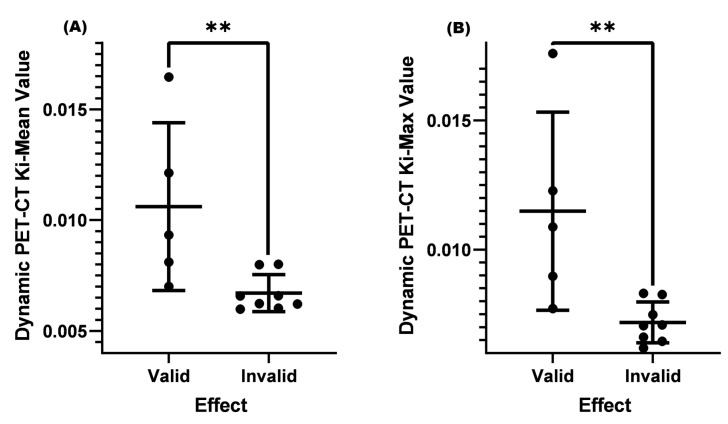
Comparison of K_i_-Mean and K_i_-Max values between the valid and invalid groups with an SUV-Max ≤ 4.5 and a lymph node < 1.0 cm and a normal EBV-DNA replication before treatment. (**A**) Comparison of K_i_-Mean between the valid and invalid groups when the SUV-Max was ≤4.5 and the cervical lymph node was <1.0 cm and there was the normal EBV-DNA replication before treatment (*p* = 0.0062 < 0.01). (**B**) Comparison of K_i_-Max between the valid and invalid groups when the SUV-Max was ≤4.5 and the cervical lymph node was <1.0 cm and there was the normal EBV-DNA replication before treatment (*p* = 0.0062 < 0.01). ** means *p* < 0.01.

**Figure 7 diagnostics-13-02530-f007:**
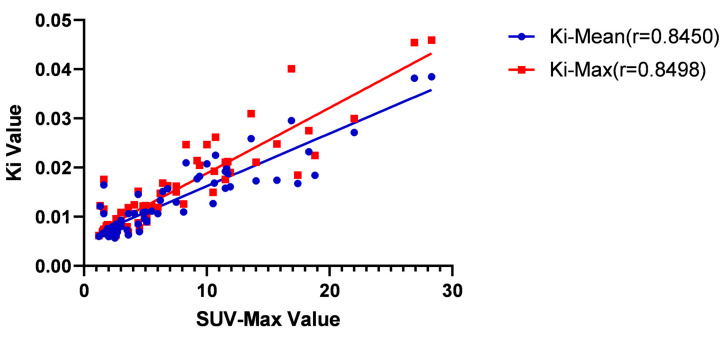
Correlation between SUV-Max and K_i_-Mean values and K_i_-Max. The r coefficient of the SUV Max and K_i_-Mean was 0.8450 (*p* < 0.01), and the r coefficient of the K_i_-Max was 0.8498 (*p* < 0.01).

**Figure 8 diagnostics-13-02530-f008:**
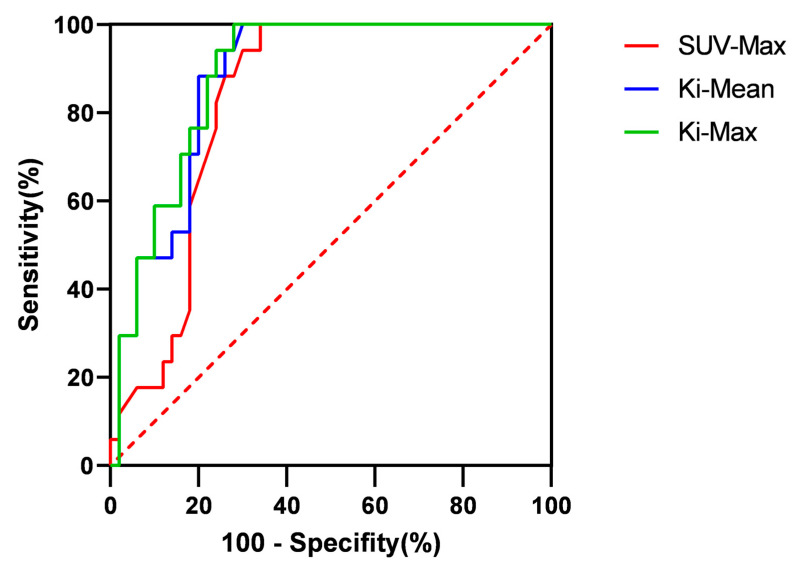
The ROC curves of SUV-Max, K_i_-Mean, and K_i_-Max.

**Table 1 diagnostics-13-02530-t001:** The characteristics of NPC patients.

Variable	Number (%)
Median Age(years)	48
Sex	
Male	36 (71)
Female	15 (29)
T Stage	
T1	7 (14)
T2	9 (17)
T3	28 (55)
T4	7 (14)
N Stage	
N1	21 (41)
N2	27 (53)
N3	3 (6)
M Stage	
Mx	3 (6)
M0	43 (86)
M1	3 (8)
Clinical Stage	
I	0 (0)
II	8 (16)
III	27 (53)
IV	15 (29)
Not Identified	1 (2)
EBV-DNA Status	
High Level	23 (45)
Normal Level	28 (55)

**Table 2 diagnostics-13-02530-t002:** Comparison of SUV-Max, K_i_-Mean, and K_i_-Max between the valid and invalid groups.

Clinical Efficacy	Number (%)	Mean SUV-Max	*p* Value	Mean K_i_-Mean	*p* Value	Mean K_i_-Max	*p* Value
Valid	50 (75)	7.2	<0.001	0.01323	<0.001	0.01510	<0.001
Invalid	17 (25)	4.3		0.00978		0.01077	

**Table 3 diagnostics-13-02530-t003:** Comparison of K_i_-Mean and K_i_-Max values between valid and invalid groups when the SUV-Max was ≤4.5.

Clinical Efficacy	Mean K_i_-Mean	*p* Value	Mean K_i_-Max	*p* Value
Valid	0.00897	>0.05	0.00982	>0.05
Invalid	0.00737		0.00787	

**Table 4 diagnostics-13-02530-t004:** Comparison of K_i_-Mean and K_i_-Max values between the valid and invalid groups with an SUV-Max ≤ 4.5 and a lymph node < 1.0 cm before treatment.

Clinical Efficacy	Mean K_i_-Mean Value	*p* Value	Mean K_i_-Max Value	*p* Value
Valid	0.00910	0.0457	0.01004	0.0298
Invalid	0.00716		0.00767	

**Table 5 diagnostics-13-02530-t005:** Comparison of K_i_-Mean and K_i_-Max values between the valid and invalid groups with an SUV-Max ≤ 4.5 and a lymph node < 1.0 cm and a normal EBV-DNA replication before treatment.

Clinical Efficacy	Mean K_i_-Mean Value	*p* Value	Mean K_i_-Max Value	*p* Value
Valid	0.01060	0.0062	0.01149	0.0062
Invalid	0.00670		0.00719	

**Table 6 diagnostics-13-02530-t006:** Correlation between different factors and K_i_-Mean values and K_i_-Max.

Factors	K_i_-Mean		K_i_-Max	
	r	*p* Value	r	*p* Value
SUV-Max	0.8450	<0.01	0.8498	<0.01
T-Stage	0.0659	0.60	0.0517	0.68
Normal EBV-DNA level	0.2171	0.08	0.2318	0.06
Age	0.0558	0.65	0.0739	0.55
Lymph node < 1.0 cm	0.6369	<0.01	0.6416	<0.01

## Data Availability

All data have been included in this article.

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
