# Peer review of "Diagnostic Value of Dynamic 18F-Fluorodeoxyglucose Positron Emission Tomography-Computed Tomography (18F-FDG PET-CT) in Cervical Lymph Node Metastasis of Nasopharyngeal Cancer"

_diagnostics, 2023, doi:10.3390/diagnostics13152530_

Round 1

Reviewer 1 Report

1- The authors should explain more about the proposed method

2- The bock diagram of the proposed method should be shown.

3- Some references are too old.

4- Conclusion section should be written.

a

Author Response

Response to Reviewer 1 Comments

Point 1: The authors should explain more about the proposed method.

Response 1: Thank you for your opinion. I have provided the more detailed explanation in the manuscript about the prosposed method.

Point 2: The bock diagram of the proposed method should be shown.

Response 2: I have added the block diagram in the manuscript.

Point 3: Some references are too old.

Response 3: I have made modifications according to your opinon. Thank you!

Point 4: Conclusion section should be written.

Response 4: I have added it in the manuscript. Thank you for your opinion.

Reviewer 2 Report

The manuscript regards a very interesting study evaluating the diagnostic value of dynamic FDG PET/CT for characterizing lymph nodes in nasopharyngeal cancer. This manuscript is well written and very clear. Authors clearly demonstrated the interest of dynamic PET /CT for defining the malignancy of lymph nodes; especially for those with a small size.

 Some very minor points can be discussed:

-       “Valid” and “unvalid” groups:  must be defined in the abstract. Due to the methodology, it is maybe not appropriate to change these terms by malignant vs no malignant groups (but better understanding).

-       P2 line 85-86: all patients had a lymph node biopsy for confirming malignancy? It is not clear

-       P3 line 119: Please define Ki mean and Ki max

-       P3 line 132: “ a reduction > 50%.....”. Is it the volume? The largest axis? The shortest axis?

-       P 9 lines 267-271: SUV max, ki mean and Ki max give very similar results. There is no superiority for one parameter and no statistical test for demonstrating superiority. 

-       P 12 lines 353-35.  AUCs are not significantly between parameters. You must use a DeLong test for comparing ROC curves and demonstrate a superiority ( but it would not be significant here). No significant differences regarding Se and Sp as well. You can speak of tendency for better accuracy using dynamic PET/CT (to be confirmed in a larger data).

Author Response

Point 1: “Valid” and “invalid” groups:  must be defined in the abstract. Due to the methodology, it is maybe not appropriate to change these terms by malignant vs no malignant groups (but better understanding).

Response 1: I have made modifications based on your feedback.

Point 2: P2 line 85-86: all patients had a lymph node biopsy for confirming malignancy? It is not clear.

Response 2: We did not perform cervical lymph node biopsies on every patient. On the one hand, when the patient suspects the nasopharyngeal malignancy on magnetic resonance imaging, but multiple endoscopic biopsies could not identify the malignancy, we found some suspicious lymph nodes for biopsies. If both malignancy and nasopharyngeal origin were considered, we identify the lymph node as cervical lymph node metastasis of nasopharyngeal carcinoma. On the other hand, for some patients who had large lumps in the neck, we also performed cervical lymph nodes biopsies after seeking the patient's consent. For most patients, due to the ability of nasal endoscopy to identify nasopharyngeal carcinoma, technical limitations, and the inability of most patients to withstand invasive procedures, the malignancy of their cervical lymph nodes is mainly determined by the degree of changes in magnetic resonance imaging before and after treatment.

Point 3: P3 line 119: Please define Ki mean and Ki max。

Response 3: I have made modifications based on your feedback.

Point 4: P3 line 132: “ a reduction > 50%.....”. Is it the volume? The largest axis? The shortest axis?

Response 4: We have deleted the paragraph. This content was described in detail in the ‘Efficacy evaluation’. We mainly measured the maximum diameter and maximum vertical diameter of the lymph node and calculated the changes before and after treatment. According to the WHO efficacy evaluation guidelines, if the change before and after treatment was≥50%, it was generally considered as ‘partical remission’ or ‘complete remission’, indicating that the treatment was effective. We considered it as the malignant lymph node with a high possibility, and if there was almost no change or no significant change before and after treatment, we consider it as biased towards benign.

Point 5: P 9 lines 267-271: SUV max, ki mean and Ki max give very similar results. There is no superiority for one parameter and no statistical test for demonstrating superiority.

Response 5: Firstly, thank you for your careful guidance. We conducted a reanalysis using SPSS 27.0 software to compare the AUC values of Ki-Mean and Ki-Max with the AUC value of SUVmax. It was found that there was no significant difference in AUC values between Ki-Mean and SUV-Max (P=0.220>0.05), and there was also no significant difference in AUC values between Ki-Max and SUVmax (P=0.159).

Point 6: P 12 lines 353-35.  AUCs are not significantly between parameters. You must use a DeLong test for comparing ROC curves and demonstrate a superiority ( but it would not be significant here). No significant differences regarding Se and Sp as well. You can speak of tendency for better accuracy using dynamic PET/CT (to be confirmed in a larger data).

Response 6: Through analysis, it was found that the AUC values, Sensitivities and specificities of Ki -Mean and Ki-Max were not significantly different from those of SUV-Max (P>0.05). We have adopted your suggestions and made modifications accordingly. Thank you for your guidance.

Reviewer 3 Report

This manuscript entitled “Diagnostic value of dynamic 18F-fluorodeoxyglucose positron emission tomography-computed tomography (18F-FDG PET-CT) in cervical lymph node metastasis of nasopharyngeal cancer” by G. L. Li et al., reveals the diagnostic value of dynamic PET-CT in cervical lymph node metastasis of nasopharyngeal cancer through analysis, and its sensitivity and accuracy are superior to static PET-CT. Several questions are suggested for authors to address in the revision.

1. The introduction of the manuscript is not complete enough, and the authors should provide the latest research progress and a detailed introduction.

2. Please supplement the captions of Figs. 5 to 7.

3. There should be a space between the number and the unit, and the letters in the unit should be in upright letters. The authors are requested to make corresponding amendments in the revision.

4. The format of the references needs to be further improved.

Minor editing of English language required.

Author Response

Point 1: The introduction of the manuscript is not complete enough, and the authors should provide the latest research progress and a detailed introduction.

Response 1: Thank you for your opinion. I have added and modified it according to your opinion.

Point 2: Please supplement the captions of Figs. 5 to 7.

Response 2: Thank you for your opinion. I have made modifications.

Point 3: There should be a space between the number and the unit, and the letters in the unit should be in upright letters. The authors are requested to make corresponding amendments in the revision.

Response 3: Thank you for your opinion. I have made modifications.

Point 4: The format of the references needs to be further improved.

Response 4: Thank you for your opinion. I have made modifications.